# On a Hidden Property in Computational Imaging

## Abstract

Computational imaging plays a vital role in various scientific and medical applications, such as Full Waveform Inversion (FWI), Computed Tomography (CT), and Electromagnetic (EM) inversion. These methods address inverse problems by reconstructing physical properties (e.g., the acoustic velocity map in FWI) from measurement data (e.g., seismic waveform data in FWI), where both modalities are governed by complex mathematical equations. In this paper, we empirically demonstrate that despite their differing governing equations, three inverse problems—FWI, CT, and EM inversion—share a hidden property within their latent spaces. Specifically, using FWI as an example, we show that both modalities (the velocity map and seismic waveform data) follow the same set of one-way wave equations in the latent space, yet have distinct initial conditions that are linearly correlated. This suggests that after projection into the latent embedding space, the two modalities correspond to different solutions of the same equation, connected through their initial conditions. Our experiments confirm that this hidden property is consistent across all three imaging problems, providing a novel perspective for understanding these computational imaging tasks.

## 1 Introduction

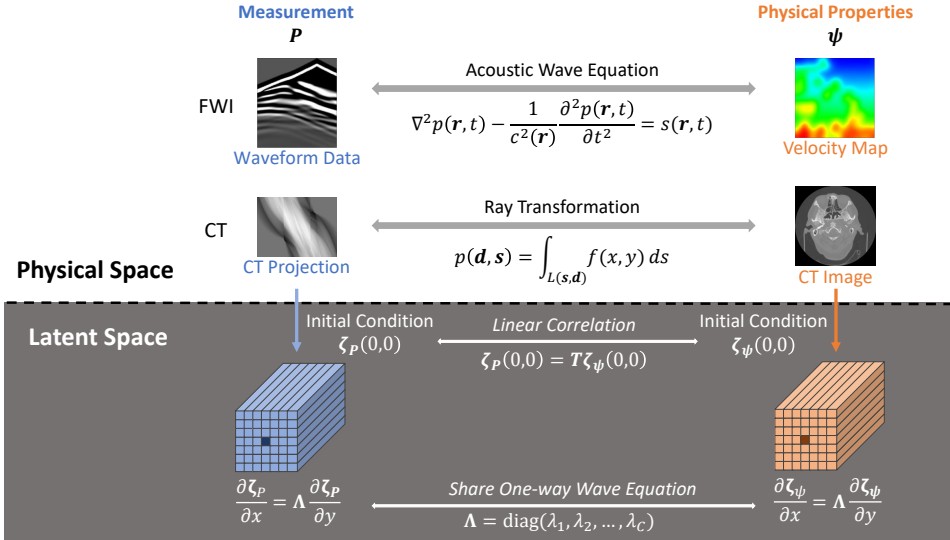

Figure 1: **Illustration of the hidden property**. Different imaging problems share a common hidden property in the latent space: the two modalities involved in each problem follow the same set of one-way wave equations in the latent space, with different but linearly correlated initial conditions. For instance, CT projection data $p(\mathbf{d}, \mathbf{s})$ and CT image $f(x, y)$, once projected into the latent space, become two distinct but linearly correlated initial conditions of the same wave equation $\frac{\partial \boldsymbol{\zeta}}{\partial x} = \boldsymbol{\Lambda} \frac{\partial \boldsymbol{\zeta}}{\partial y}$.

Computational imaging, encompassing applications such as Full Waveform Inversion (FWI), Computed Tomography (CT), and Electromagnetic (EM) inversion, is foundational in many scientific

and medical fields. These methods address inverse problems, which involve reconstructing physical properties from measured data, a process governed by linear or nonlinear mathematical equations (Kirsch et al., 2011). Accurate reconstruction of physical properties is essential for various applications, including medical diagnostics, geophysical exploration, and non-destructive testing of materials. Deep learning methods usually trade these problems as Image-to-Image translation tasks, modeling them via encoder-decoder architectures, and achieve significant improvements (McCann et al., 2017; Wu & Lin, 2019; Ongie et al., 2020; Song et al., 2022; Deng et al., 2022; Jin et al., 2022; Feng et al., 2024b). However, while these methods construct latent space representations, typically with a bottleneck in the network, they lack a deeper understanding of these latent representations. Thus, we are curious about the question:

> *Whether an elegant mathematical relationship exists in the latent space, akin to that in the original space*?

This curiosity drives us to explore the structure of the latent space, specifically whether a simpler mathematical relationship exists between the two modalities in these inverse problems.

Recently, Chen et al. demonstrated that, in the latent space, natural images can be described by a set of one-way wave equations with learnable speeds (Chen et al., 2023b;a), where each image corresponds to a unique solution of these wave equations, enabling high-fidelity reconstruction from an initial condition. While this work links natural images to wave equation-based representations, it is limited to single-modality image reconstruction. Motivated by this work, we aim to explore the relationship between two modalities in computational imaging. Specifically, our exploration is driven by three key questions: (1) Can two modalities share the same wave equations in the latent space? (2) What is the relationship between their initial conditions? (3) Can this relationship generalize across different computational imaging problems?

This paper answers these three questions above. Firstly, we show that the latent spaces of both measurement data and target properties are governed by the same set of one-way wave equations, characterized by identical wave speeds. The two modalities can be projected as different initial conditions of these same equations. Secondly, building upon the work of Feng et al. (Feng et al., 2022; 2024b), who discovered a linear correlation between the latent representations of two modalities in geophysical inversion problems (e.g., FWI, EM inversion), we further reveal that when the two modalities follow the same wave equations, the corresponding initial conditions also exhibit a strong linear correlation, allowing one to be derived from the other via a linear transformation. Finally, we demonstrate that this hidden property is common across different computational imaging problems. As illustrated in Fig 1, we term this hidden property HINT (short for the **HI**dde**N** proper**T**y). The HINT transforms the relationships of physical properties, traditionally described by distinct equations in the physical space, into a dual problem in the latent space described by this common property across various tasks.

The proposed hidden property can be easily implemented. We propose a unified framework that learns the embedding of measurement data and target property together while simultaneously generating input reconstruction and target property prediction. Our approach begins by encoding the measurement data $P$ (e.g., waveform data in FWI) into a latent vector, denoted as $v_P$, using a visual encoder $\mathcal{E}$. This latent vector $v_P$ is then linearly transformed to obtain the latent vector $v_\psi$ of the target property $\psi$ (e.g., velocity map in FWI). Both $v_P$ and $v_\psi$ are propagated through the same autoregression process (called multi-path FINOLA) governed by one-way wave equations (Chen et al., 2023b;a) to generate larger size feature maps $z_P$ and $z_\psi$, respectively. Subsequently, decoders $\mathcal{D}_P$ and $\mathcal{D}_\psi$ are employed to reconstruct the original input $P$ from $z_P$ and to infer the corresponding $\psi$ from $z_\psi$. The network is trained with a combination of $L_1$ and $L_2$ loss. This integrated framework captures both cross-domain and within-domain relationships in the latent space, offering a more precise and interpretable understanding of the latent space structure. The discovered hidden property forms the core of the framework, serving as a hard constraint throughout the learning process. Based on this architecture, the wave speed $\Lambda$ of the hidden wave equations, along with the two solutions (noted as $\zeta_P$ and $\zeta_\psi$), can be derived from the parameters of FINOLA, as well as the feature maps $z_P$ and $z_\psi$. The detailed relationship will be explained in the next section.

We validate the proposed hidden property across three tasks: FWI (Deng et al., 2022), EM inversion (Alumbaugh et al., 2021), and CT (Flanders et al., 2020). Across these tasks, our approach matches or surpasses the performance of unconstrained methods. These results demonstrate that the constrained latent space remains optimal for solving inverse problems, offering a simpler and more

tractable latent space structure without compromising reconstruction accuracy. By leveraging the hidden property, the proposed framework provides a new perspective on the relationship between physical properties in their latent representations, paving the way for a further understanding of the latent space.

## 2 THE HIDDEN PROPERTY

In this section, we provide a detailed introduction to the hidden property. First, we review three computational imaging tasks, each involving predicting one modality (physical property) from another modality (measurement data). Next, we demonstrate how to extend FINOLA from one modality to two modalities that share the same one-way wave equations in the latent space and illustrate the implementation details. Finally, we formally summarize the proposed hidden property.

### 2.1 REVIEW OF COMPUTATIONAL IMAGING TASKS

**Full waveform inversion (FWI)** is a well-known method to infer subsurface acoustic velocity maps from seismic waveform data. Specifically, seismic waveform data are collected via seismic surveys, during which receivers record reflected and refracted seismic waves generated by controlled sources. Each receiver logs a 1D time series signal, and the collective signals from all receivers form the waveform data. Let $p(\boldsymbol{r}, t)$ represent the waveform data, and $c(\boldsymbol{r})$ is the velocity map. $s(\boldsymbol{r}, t)$ is the source term. $\boldsymbol{r} = (x, y)$ is the spatial location for 2D slice data, in which $x$ is the horizontal direction and $z$ is the depth, $t$ denotes time, and $\nabla^2$ is the Laplacian operator. The process is mathematically governed by the acoustic wave equation:

$$\nabla^2 p(\boldsymbol{r}, t) - \frac{1}{c^2(\boldsymbol{r})} \frac{\partial^2}{\partial t^2} p(\boldsymbol{r}, t) = s(\boldsymbol{r}, t). \tag{1}$$

In this task, the aim is to predict the velocity map $c(\boldsymbol{r})$ (i.e., target property $\psi$) from the waveform data collected by surface sensors (i.e., $z = 0$), abbreviated as $p(x, t)$ (i.e., measurement data $\boldsymbol{P}$).

**Computed Tomography (CT)** is a vital imaging technique used to capture cross-sectional images of an object's internal structure. In CT, X-rays are passed through the object at various angles, and the resulting attenuation is measured as projection data. Let $f(x, y)$ represent the internal structure (i.e., the attenuation coefficient), where $(x, y)$ are the spatial coordinates. The projection data $p(\mathbf{d}, \mathbf{s})$ is a function of the X-ray source position $\mathbf{s} = (x_s, y_s)$ and detector position $\mathbf{d} = (x_d, y_d)$, measuring the total X-ray attenuation along the path between the source and detector. Let $L(\mathbf{s}, \mathbf{d})$ is the line segment connecting the source $\mathbf{s}$ and the detector $\mathbf{d}$, and $ds$ is the differential element along this line. Mathematically, the projection data is expressed as:

$$p(\mathbf{d}, \mathbf{s}) = \int_{L(\mathbf{s}, \mathbf{d})} f(x, y) \, ds. \tag{2}$$

In this task, the aim is to predict attenuation image $f(x, y)$ (i.e., target property $\psi$) from the projection data $p(\mathbf{d}, \mathbf{s})$ (i.e., measurement data $\boldsymbol{P}$).

**Electromagnetic (EM) inversion** focuses on recovering subsurface conductivity from surface-acquired electromagnetic measurements. Let $\mathbf{E}$ and $\mathbf{H}$ are the electric and magnetic fields. $\mathbf{J}$ and $\mathbf{P}$ are the electric and magnetic sources. $\sigma$ is the electrical conductivity and $\mu_0 = 4\pi \times 10^{-7} \Omega \cdot s/m$ is the magnetic permeability of free space. The governing equations here are time-harmonic Maxwell's Equations

$$\sigma \mathbf{E} - \nabla \times \mathbf{H} = -\mathbf{J},$$
$$\nabla \times \mathbf{E} + i\omega \mu_0 \mathbf{H} = -\mathbf{M}. \tag{3}$$

In this task, the aim is to predict electrical conductivity $\sigma$ (i.e., target property $\psi$) from the electric and magnetic fields $\mathbf{E}$ and $\mathbf{H}$ (i.e., measurement data $\boldsymbol{P}$).

### 2.2 REVIEW OF FINOLA FOR IMAGES

**Vanilla FINOLA for one modality:** FINOLA (Chen et al., 2023b) is a First-Order Norm+Linear Autoregressive process that generates a feature map $\boldsymbol{z}(x, y)$ by predicting each position using its

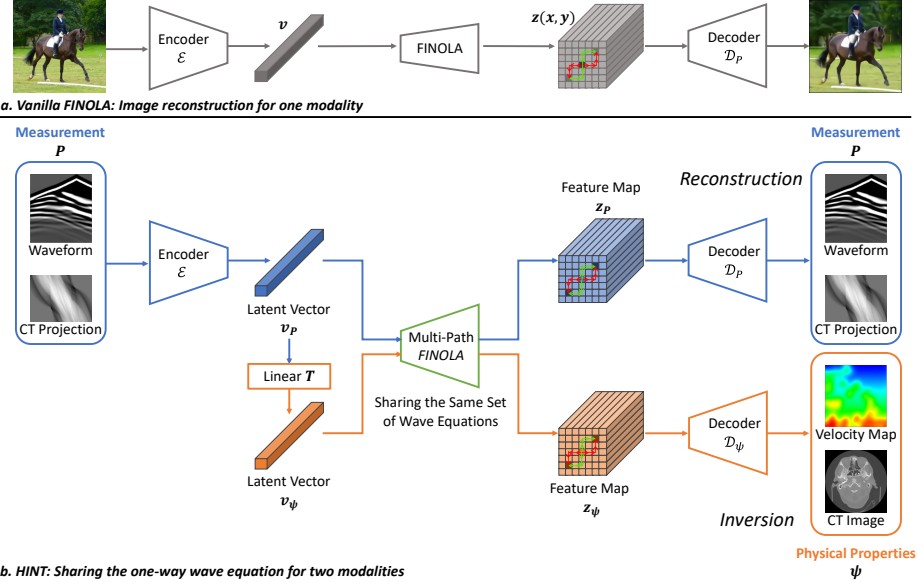

Figure 2: **Comparsion of Vanila FINOLA with the proposed HINT**. The subfigure **a) is the framework for Vanilla FINOLA**, which reconstructs images within one modality. The illustration figures are from Chen et al. (2023b). The subfigure **b) is the overview of our framework**. Each measurement $P$ is firstly encoded into a single vector $v_P$. The latent vector $v_\psi$ is then obtained from a linear transformation $T$. A shared multi-path FINOLA layer is applied to autoregress the feature map $z_P$ and $z_\psi$, respectively. Finally, two separate decoders composed of upsampling and $3 \times 3$ convolutional layers are used to reconstruct the measurement and to invert the target property.

immediate previous neighbor. An illustration of FINOLA is shown in Fig. 2 (a). It begins with encoding an image to a single vector $v$. Then, this vector will be used as the initial condition, i.e., $z(0,0) = v$, to regress the entire feature map via the following equations recursively:

$$\frac{\partial z}{\partial x} = A\hat{z}(x,y), \quad \frac{\partial z}{\partial y} = B\hat{z}(x,y), \quad \hat{z}(x,y) = \frac{z(x,y) - \mu_z}{\sigma_z}, \tag{4}$$

where the matrices $A$ and $B$ are learnable parameters with dimensions $C \times C$. $\hat{z}(x,y)$ is the normalized $z(x,y)$ over $C$ channels at position $(x,y)$. The mean $\mu_z = \frac{1}{C}\sum_k z_k(x,y)$ and the standard deviation $\sigma_z = \sqrt{\sum_k (z_k(x,y) - \mu_z)^2 / C}$ are calculated at each position $(x,y)$ over $C$ channels. Finally, a lightweight decoder is used to reconstruct the image.

**Hidden wave explanation:** The hidden waves phenomenon (Chen et al., 2023a) provides a new interpretation of FINOLA through the lens of wave equations. The term "hidden" refers to the speeds of waves that are latent but learnable. In particular, it needs to meet two conditions: (a) the matrix $B$ is invertible, and (b) the matrix $AB^{-1} = V\Lambda V^{-1}$ is diagonalizable, where $V$ constitute a basis of eigenvectors and $\Lambda$ represent the corresponding eigenvalues, i.e., $\Lambda = diag(\lambda_1, \lambda_2, ..., \lambda_C)$. Then, let $\zeta = V^{-1}z$, the Eq. 4 can be simplified as

$$\frac{\partial \zeta}{\partial x} = \Lambda \frac{\partial \zeta}{\partial y}, \tag{5}$$

where each dimension of $\zeta$ follows a one-way wave equation, with initial condition $\zeta(0,0) = V^{-1}v$. Typically, the one-way wave equation involves time $t$; here, it is replaced by $y$. This formulation allows each image to correspond to a solution of the one-way wave equations.

## 2.3 EXTENDING FINOLA TO TWO MODALITIES

In the subsection, we use FWI as an example to illustrate how to extend FINOLA to two modalities (waveform data and velocity map). This extension can be applied to CT and EM in a straightforward manner.

**FINOLA for source modality (e.g., measurement):** The measurement data (e.g., waveform data) follows a similar process as vanilla FINOLA, illustrated by the blue arrow in Fig.2 (b). First, the

measurement data $\boldsymbol{P}$ is encoded into into a latent vector $\boldsymbol{v_P} = \mathcal{E}(\boldsymbol{P})$ with a Transformer encoder $\mathcal{E}$. An attention pooling (Lee et al., 2019; Yu et al., 2022; Chen et al., 2023b) is applied in the last layer of the encoder to obtain the compressed vector. Then, used as the initial condition, $\boldsymbol{v_P}$ is propagated through a FINOLA layer to generate a larger feature map $\boldsymbol{z_P}$. Mathematically, it is represented as:

$$\frac{\partial \boldsymbol{z_P}}{\partial x} = \boldsymbol{A}\hat{\boldsymbol{z}}_{\boldsymbol{P}}(x, y), \quad \frac{\partial \boldsymbol{z_P}}{\partial y} = \boldsymbol{B}\hat{\boldsymbol{z}}_{\boldsymbol{P}}(x, y). \tag{6}$$

In practice, we apply the multi-path FINOLA implementation, which divides the initial conditions into multiple vectors, with each vector subjected to the FINOLA process. All these paths have the same parameters. Subsequently, the resulting feature maps, each representing a special solution that satisfies the necessary constraints, are aggregated to form the final solution $\boldsymbol{z_P}$. At the end, a decoder $\mathcal{D}_P$ is then employed to reconstruct the original input $\boldsymbol{P} = \mathcal{D}_P(\boldsymbol{z_P})$. The decoder is designed with a series of upsampling layers followed by $3 \times 3$ convolutional layers equipped with residual connections.

**FINOLA for target modality (e.g., physical property):** To deal with two modalities in computation imaging, we extend FINOLA to incorporate two modalities and force them to share FINOLA parameters. It is shown in the orange arrow in Fig. 2 (b). To produce the latent vector $\boldsymbol{v}_{\boldsymbol{\psi}}$, which corresponds to the target property $\psi$ (i.e., velocity map), $\boldsymbol{v_P}$ is linearly transformed, with the linear lay $\boldsymbol{T}$. Note that both vectors have the same dimensionality. Then, $\boldsymbol{v}_{\boldsymbol{\psi}}$ is propagated through the same FINOLA layer to generate the feature map $\boldsymbol{z}_{\boldsymbol{\psi}}$. Mathematically, this is represented as:

$$\boldsymbol{v}_{\boldsymbol{\psi}} = \boldsymbol{T}\boldsymbol{v_P} \quad \frac{\partial \boldsymbol{z}_{\boldsymbol{\psi}}}{\partial x} = \boldsymbol{A}\hat{\boldsymbol{z}}_{\boldsymbol{\psi}}(x, y), \quad \frac{\partial \boldsymbol{z}_{\boldsymbol{\psi}}}{\partial y} = \boldsymbol{B}\hat{\boldsymbol{z}}_{\boldsymbol{\psi}}(x, y), \tag{7}$$

where the matries $\boldsymbol{A}$ and $\boldsymbol{B}$ are shared across two modalities. To evaluate the quality of the latent space, another convolutional decoder $\mathcal{D}_\psi$ is employed to infer the target property $\psi = \mathcal{D}_\psi(\boldsymbol{z}_{\boldsymbol{\psi}})$.

**Overall Structure:** Combining the above two processes over two modalities, we proposed method HINT (short for the **Hi**dden **Pro**perty), a unified framework that jointly learns the embeddings of both measurement data and target property, while simultaneously performing input reconstruction and target property prediction. The overall framework is illustrated in Fig. 2 (b). The network is trained by combining both the reconstruction loss and prediction loss.

**Empirical validation:** We empirically validate the two key components of the above extension of two modalities: the shared wave equations and the linear correlation between embeddings. First, we compare using separate versus shared FINOLA layers on the FWI tasks. Results are shown in Fig.3, Section 3.3. We see similar performance between models using two distinct FINOLAs and those sharing one, confirming the efficiency of the shared configuration. Next, we test nonlinear converters, including Maxout and MLP, against the linear converter. Results are shown in Fig.4, Section 3.3. A nonlinear converter has no positive effect, affirming that a strong linear correlation effectively captures the relationship between the two modalities without needing complex mappings.

## 2.4 HIDDEN PROPERTIES

The empirical validation above (i.e., shared FINOLA parameters across two modalities and the linear correlation between latent vectors) reveals two hidden properties:

**Empirical Property 1: Two modalities correspond to two solutions of a common set of one-way wave equations.** Following the hidden wave explanation for FINOAL in Eq. 5, letting $\boldsymbol{AB}^{-1} = \boldsymbol{V}\boldsymbol{\Lambda}\boldsymbol{V}^{-1}$, where $\boldsymbol{\Lambda}$ is the diagonal eigenvalues, we define

$$\boldsymbol{\zeta_P} = \boldsymbol{V}^{-1}\boldsymbol{z_P}, \quad \boldsymbol{\zeta_\psi} = \boldsymbol{V}^{-1}\boldsymbol{z_\psi}. \tag{8}$$

Then, based on Eq. 6 and 7, we can extend the hidden wave to both modalities that follow the same set of one-way wave equations in the latent space, characterized by the same wave speeds $\boldsymbol{\Lambda}$:

$$\frac{\partial \boldsymbol{\zeta_P}}{\partial x} = \boldsymbol{\Lambda}\frac{\partial \boldsymbol{\zeta_P}}{\partial y}, \qquad \frac{\partial \boldsymbol{\zeta_\psi}}{\partial x} = \boldsymbol{\Lambda}\frac{\partial \boldsymbol{\zeta_\psi}}{\partial y}. \tag{9}$$

This indicates that, despite representing different physical aspects, the two modalities correspond to distinct solutions of the same set of one-way wave equations governed by the same wave dynamics.

**Empirical Property 2: The initial conditions of two modalities are linearly correlated.** With the wave equation format in Eq.9, both latent embeddings of two modalities are merely different initial conditions of the same wave equations. One can be derived from the other through a linear transformation. With the linear converter $\boldsymbol{T}$, the relationship between the two initial conditions can be formulated as

$$\zeta_{\psi}(0,0) = \boldsymbol{T}\zeta_{\boldsymbol{P}}(0,0), \tag{10}$$

where the initial conditions are computed as $\zeta_{\boldsymbol{P}}(0,0) = \boldsymbol{V}^{-1}\boldsymbol{v}_{\boldsymbol{P}}$ and $\zeta_{\psi}(0,0) = \boldsymbol{V}^{-1}\boldsymbol{v}_{\psi}$.

**Difference with vanilla FINOLA:** Unlike vanilla FINOLA, which is designed for single-modality image reconstruction, our method extends to two modalities by sharing parameters across both domains. While vanilla FINOLA captures single-domain image invariants, we use FINALO to model the relationship between two domains in computational imaging, enabling the joint representation of measurement data and target properties with wave equations.

## 3 EXPERIMENTS

In our experiments, we first examine the proposed hidden property through two key aspects: 1) the shared wave equation and 2) the linear correlation, using the FWI task as an example. We then evaluate our approach across three import computational imaging tasks, FWI, CT, and EM inversion, to demonstrate the consistency of the hidden property across different tasks. Finally, we present an ablation study of the feature map size generated via FINOLA.

### 3.1 DATASETS

**FWI:** For many scientific problems, like subsurface imaging, real data are extremely expensive and difficult to obtain. Research often relies on full-physics simulations due to the lack of publicly available real datasets. Thus, we verify our method on OpenFWI (Deng et al., 2022), the first open-source collection of large-scale, multi-structural benchmark datasets for data-driven seismic FWI. It contains 11 2D datasets with baseline, which can be divided into four groups: four datasets in the "Vel Family" are FlateVel-A/B, and CurveVel-A/B; four datasets in the "Fault Family" are FlateFault-A/B, and CurveFault-A/B; two datasets in "Style Family" are Style-A/B; and one dataset in "Kimberlina Family" is Kimberlina-$CO_2$. The first three families cover two versions: easy (A) and hard (B), in terms of the complexity of subsurface structures. The following experiments are conducted on the ten datasets of these first three families. We will use the abbreviations (e.g., FVA for FlatVel-A). More details can be found in (Deng et al., 2022).

**CT:** The CT dataset, provided by the Radiological Society of North America (RSNA) and ASNR, includes large volumes of de-identified brain CT scans labeled by expert neuroradiologists (Stein et al., 2019). It focuses on detecting acute intracranial hemorrhage, a critical condition that requires rapid diagnosis. The dataset covers various hemorrhage types to enable AI algorithms to assist in identifying hemorrhages for quicker and more accurate medical treatment. We randomly select 47000 samples as the training set and 6000 samples as the test set, with resolution $256 \times 256$. We simulate CT measurements (projection) with a stationary head CT (s-HCT) system with three linear CNT x-ray source arrays (Luo et al., 2021). This design has sparse and asymmetrical scans and a non-circular geometry with a relatively low radiation dose, providing a unique challenge to the reconstruction. An illustration of the geometry has been shown in the Supplementary Material.

**EM Inversion:** We also test our method on the subsurface electromagnetic (EM) inversion task on the Kimberlina-Reservoir dataset, which recovers subsurface conductivity from surface-acquired EM measurements. The geophysical properties were developed under DOE's NRAP. It is based on a potential $CO_2$ storage site in the Southern San Joaquin Basin of California (Alumbaugh et al., 2021). In this data, there are 780 EM data for geophysical measurement with the corresponding conductivity. We use 750/30 for training and testing. EM data are simulated by finite-difference method (Commer & Newman, 2008; Feng et al., 2022).

### 3.2 IMPLEMENTATION DETAILS

**Training Details.** The data are normalized to the range [-1, 1]. We employ AdamW (Loshchilov & Hutter, 2018) optimizer with momentum parameters $\beta_1 = 0.9$, $\beta_2 = 0.999$ and a weight decay of

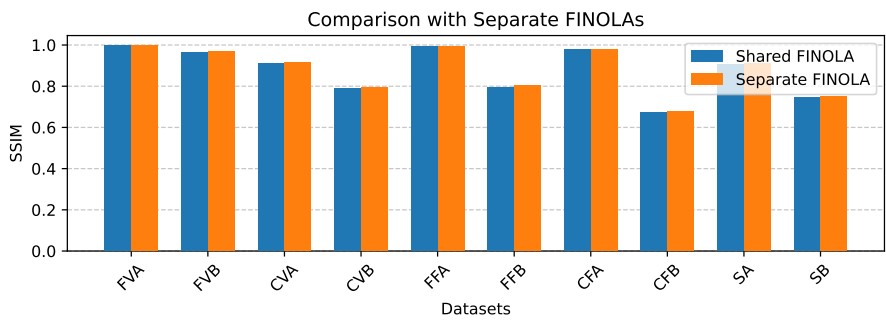

Figure 3: **Comparing HINT with a two-separate-FINOLAs network**, where each embedding has its own set of wave speeds, in terms of SSIM. Evaluated on OpenFWI.

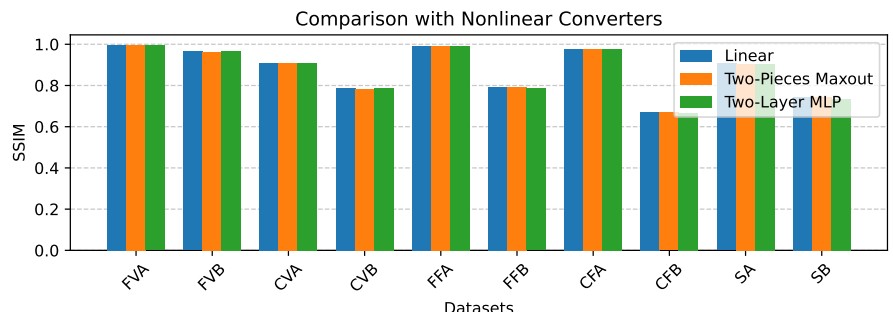

Figure 4: **Comparing HINT with nonlinear converters**, in terms of SSIM. Evaluated on Open-FWI.

0.05. The initial learning rate is set to be $1 \times 10^{-3}$, and decayed with a cosine annealing (Loshchilov & Hutter, 2016). The batch size is set to 64. We use MAE plus MSE loss to train the model. We implement our models in Pytorch, training on 8 NVIDIA Tesla V100 GPUs.

**Architecture Details:** For datasets in OpenFWI, the size of waveform data is $5 \times 1000 \times 70$, and the size of velocity maps is $70 \times 70$. We choose patch size $(100 \times 10)$ for the three-layer Transformer encoder with the hidden size of 512, and the number of heads is 16. The feature map $\zeta_P$ will be recovered to the same size as the encoder's outputs before pooling (i.e., $10 \times 7$). We use the FINOLA with a dimension of 512 and one path. The feature map of velocity maps, $\zeta_\psi$, has the size $(7 \times 7)$ for Sec. 3.3, and $(14 \times 14)$ for the rest.

For the CT dataset, the size of projection data is $3 \times 45 \times 1728$, and the size of the CT image is $256 \times 256$. We choose patch size $(9 \times 36)$ for the three-layer Transformer with the hidden size of 768, and the number of heads is 16. Then it will be pooled with two seeds, i.e., the dimension of $v_P$ is 1536. For this larger dimension, we use the FINOLA with dimension 192 in the 8 paths. The feature map $\zeta_P$ will be recovered to the same size as the encoder's outputs before pooling (i.e., $5 \times 48$). The feature map, $\zeta_\psi$, has the size $(32 \times 32)$.

**Evaluation Metrics.** We apply three metrics to evaluate the generated geophysical properties: MAE, MSE, and Structural Similarity (SSIM). Following the existing literature (Wu & Lin, 2019; Feng et al., 2022; Deng et al., 2022), MAE and MSE are employed to measure the pixel-wise error, and SSIM is to measure the perceptual similarity since the target properties have highly structured information, and degradation or distortion can be easily perceived by a human. We calculate them on normalized data, i.e., MAE and MSE in the scale $[-1, 1]$, and SSIM in the scale $[0, 1]$.

### 3.3 INSPECTION OF THE HIDDEN PROPERTY

In this part, we validate two key components of our hidden property: the shared set of wave equations and the linear correlation between two embeddings. We test them one by one to assess how well they hold in maintaining the quality of latent representations, which impacts the overall performance.

**Shared wave speed V.S. Separate wave speed.** We conducted experiments to compare the model using two separate sets of wave speeds with our approach, which shares a single set of wave speeds

across all ten datasets in the OpenFWI dataset. The SSIM for both methods is presented in Fig. 3. Models using two distinct FINOLAs exhibited similar performance, with differences being less than 1%. The results demonstrate that the latent representations produced by the shared FINOLA are of comparable quality to those generated by using two separate FINOLAs, validating the effectiveness of the proposed property. These findings confirm that the two latent representations share the same set of wave speeds without compromising the model's effectiveness.

**Linear Converter V.S. Non-Linear Converter.** We evaluate networks with more complicated non-linear converters on OpenFWI. We test a two-piece Maxout and a two-layer MLP. The results are provided in Fig 4. As the results indicate, the nonlinear mapping performs at a similar level to the linear converter, showing no overall positive effect on final performance. This outcome aligns with our conclusion that a strong linear correlation is sufficient to capture the underlying relationships between the embedding of two modalities.

## 3.4 VALIDATION ACROSS MULTIPLE COMPUTATIONAL IMAGING TASKS

**FWI:** To demonstrate the broad applicability of the hidden property, we train our model across all ten datasets in OpenFWI together. Fig 6 shows the comparison results with Inversion-Net (Wu & Lin, 2019) and Auto-Linear (Feng et al., 2024a). For a fair comparison, we used the BigFWI version of InversionNet (Jin et al., 2024), which is also trained on all ten datasets. Our model delivers overall performance that is generally similar to BigFWI, though slightly better. However, it only has three-quarters of the model size (18.2M related to inversion vs. 24.4M). It consistently outperforms Auto-Linear in all three metrics. Detailed quantitative results are available in the Supplementary Material. Figure 5 illustrates the velocity maps inverted by each method. From the figure, we can observe: 1) Our model's superior performance is reflected not only in the quantitative results but also in the visual quality of the results; 2) On certain datasets (e.g., CFB), patterns from other datasets seem to influence the results, which could indicate a limitation in how the model handles dataset-specific features when trained jointly across multiple datasets.

In Table 1, we show the reconstruction error of our model. The low reconstruction error, along with the high inverse accuracy, proves that the hidden property holds that the same set of wave equations can be shared for two embeddings. Abvoe's two experiments show that, for the set of wave equations in the latent space, the wave speed can not only be shared across embeddings of different physical quantities but can also be shared across datasets with very different subsurface structures.

**CT:** For the CT task, we choose simultaneous iterative reconstruction techniques (SIRT) (Van Aarle et al., 2016) and a modified InversionNet as the baselines. For the modified InversionNet, we double the network dimension with a deeper decoder to fit the larger CT data.

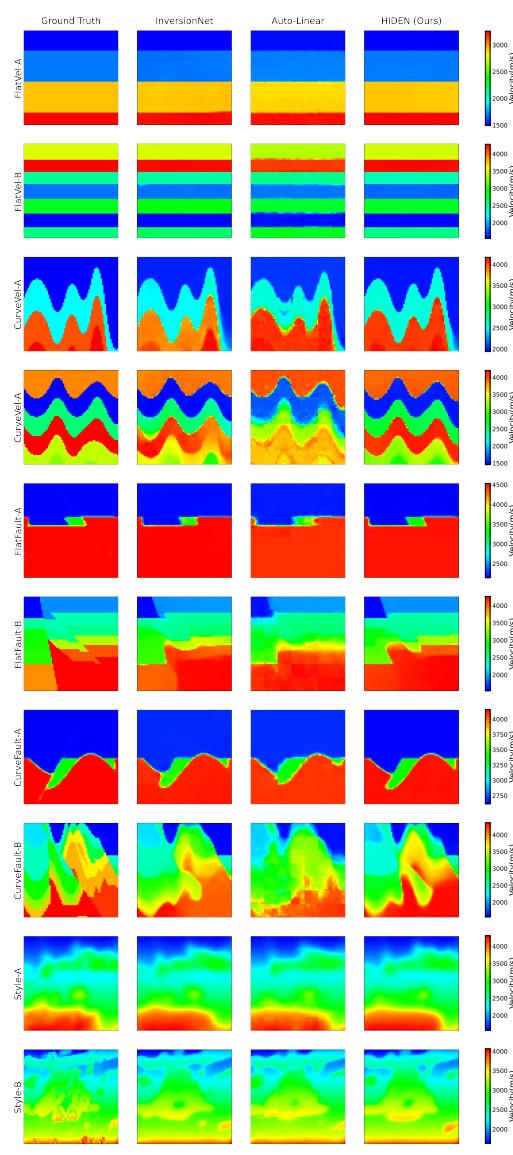

Figure 5: **Illustration of results on OpenFWI**, compared with InversionNet and Auto-Linear.

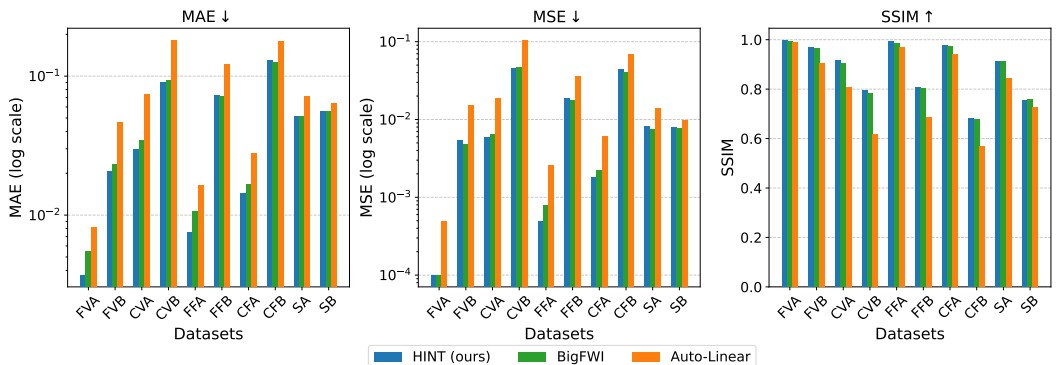

Figure 6: **Results for FWI**, compared with BigFWI and Auto-Linear for MAE, MSE, and SSIM.

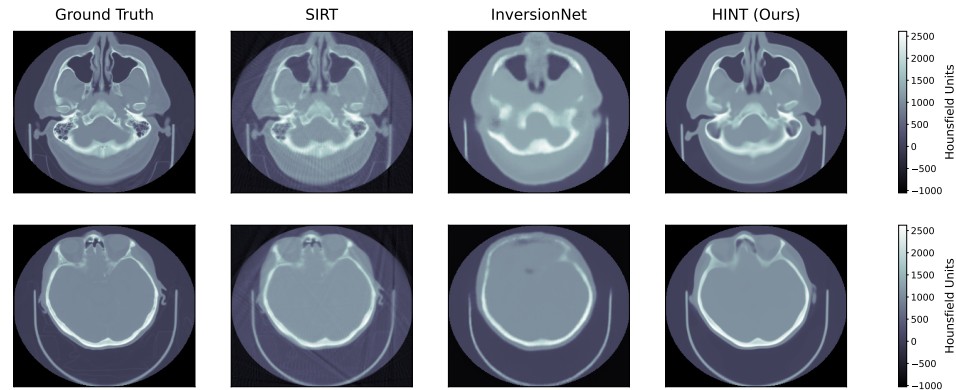

Figure 7: **Illustration of results on RSNA for CT**, compared with InversionNet and SIRT.

Table 2 shows the results of prediction. HINT outperforms InversionNet in all three metrics, demonstrating its enhanced ability to manage the complex structure of CT data. While SIRT achieves the lowest MSE, HINT delivers the best MAE, suggesting that the hidden property holds in CT data as well. Figure 7 illustrates the CT images inferred by each method. The figure shows that our model produces smoother results, which may lack some fine details. In contrast, SIRT retains more detail but introduces noticeable artifacts. Each method has its advantages, with our approach providing cleaner reconstructions and SIRT capturing more structural information at the cost of increased noise. The poor performance of InversionNnet and the comparable performance between HINT and SIRT also highlight the challenges posed by the specific CT geometry with sparse and asymmetrical scans and relatively low radiation dose.

Table 1: Quantitative results of waveform data reconstruction on OpenFWI.

| Metric | FVA | FVB | CVA | CVB | FFA | FFB | CFA | CFB | SA | SB |
|---|---|---|---|---|---|---|---|---|---|---|
| MAE↓ | 0.0014 | 0.0059 | 0.0088 | 0.0195 | 0.0031 | 0.0122 | 0.0052 | 0.0188 | 0.0050 | 0.0089 |
| MSE↓ | 1.09e-5 | 0.0001 | 0.0003 | 0.0013 | 6.96e-5 | 0.0007 | 0.0002 | 0.0012 | 0.0001 | 0.0003 |
| SSIM↑ | 0.9998 | 0.9981 | 0.9879 | 0.9757 | 0.9978 | 0.9783 | 0.9953 | 0.9585 | 0.9967 | 0.9867 |

**EM Inversion:** For the EM Inversion task, we also compare our method with InversionNet and Auto-Linear. Table 3 shows the results. Note that, to maintain consistency with previous works (Feng et al., 2024a), the MAE and MSE reported below were calculated after denormalizing to the original range of $[0, 0.65]$. We observe that our proposed HINT yields much better performance than those obtained using Auto-Linear and InversionNet. These results demonstrate that the discovered hidden property is consistent across various computational imaging tasks.

## 3.5 VALIDATION ACROSS DIFFERENT RESOLUTIONS

In this ablation, We empirically validate the wave equations by assessing HINT's performance across various feature map resolutions. Fig. 8 displays SSIM across different feature map resolutions evaluated on OpenFWI. The performance remains consistent across most resolutions, with slightly reduced performance at $35 \times 35$. This decrease is primarily due to a significantly shallow decoder. The

Table 2: **Quantitative results for CT**. MAE and MSE are calculated after denormalizing to their original range ($[-1000, 32700]$)

| Model | MAE↓ | MSE↓ | SSIM↑ |
|---|---|---|---|
| HINT | **31.95** | 9754.48 | 0.9843 |
| InversionNet | 63.27 | 274350.78 | 0.9684 |
| SIRT | 45.67 | **6510.67** | **0.9918** |

Table 3: **Quantitative results for EM inversion**. MAE and MSE are calculated after denormalizing to their original range ($[0, 0.65]$).

| Model | MAE↓ | MSE↓ | SSIM↑ |
|---|---|---|---|
| HINT | **0.0018** | **3.34e-5** | **0.9937** |
| Auto-Linear | 0.0044 | 1.92e-4 | 0.9700 |
| InversionNet | 0.0133 | 8.55e-4 | 0.9175 |

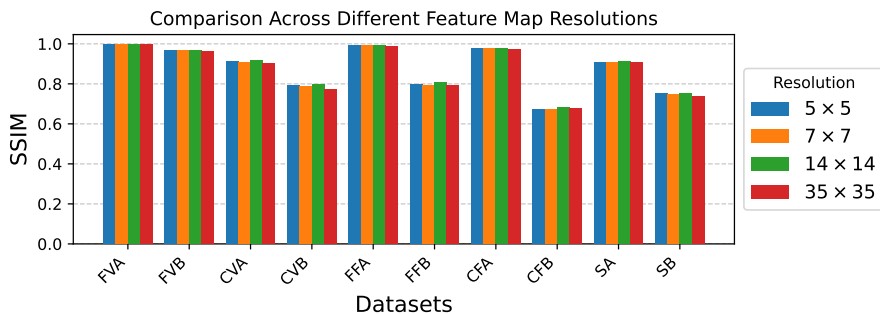

Figure 8: **Validation across multiple $z_\psi$ resolutions**, in terms of SSIM. Evaluated on OpenFWI.

quantitative results are shown in the Supplementary Material. These results demonstrate that the two modalities share wave equation representations consistently across different feature map resolutions (i.e., different wave propagation steps), affirming the validity of the revealed hidden property.

## 4 RELATED WORKS

Recently, data-driven methods for inverse problems have emerged, treating it as an image-to-image translation problem with an encoder-decoder architecture. Wu & Lin (2019); Zhang et al. (2019) utilized a CNN to address FWI, while Jin et al. (2022) combined forward modeling with deep neural networks in an unsupervised learning framework. Diffusion models have also emerged as competitive solutions for inverse problems, requiring pre-training of a prior model and integrating the measurement process into the denoising process (Song et al., 2021; Tewari et al., 2023). Unlike them, our work focuses on uncovering the underlying mathematical relationships within the latent space. Similarly, Feng et al. (2022; 2024a) decoupled the training of the encoder and decoder, demonstrating a strong linear correlation between the latent representations of two modalities in geophysical inversion. We go further by proposing that the linear correlation exists even when both modalities follow the same wave equations in the latent space.

FINOLA (Chen et al., 2023b;a), a recent advancement in modeling image invariance in latent space, models latent features using a first-order autoregressive process. It focuses on treating each image as a unique solution of the wave equations. This approach not only has the ability for image reconstruction but also extends to self-supervised learning tasks with Masked Image Modeling (MIM). In MIM (Bao et al., 2021; Xie et al., 2022), networks are challenged to reconstruct missing parts of an image. Recently, MAE (He et al., 2022) adopts an asymmetric encoder-decoder architecture to recover pixels from highly masked images, demonstrating its ability to learn robust representations. A more detailed comparison of our work with FINOLA is shown in Sec. 2

## 5 CONCLUSION

In this paper, we empirically reveal a hidden property in the latent space of computational imaging. This property, characterized by a shared set of one-way wave equations and a strong linear correlation between the latent representations of measurement data and target properties, enables a unified framework across different computational imaging tasks. Our experiments validate the hidden property across different computational imaging tasks. It shows that an elegant mathematical relationship exists in the latent space, akin to that in the original space.

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
