# A APPENDIX

## A.1 ILLUSTRATION OF S-HCT GEOMETRY

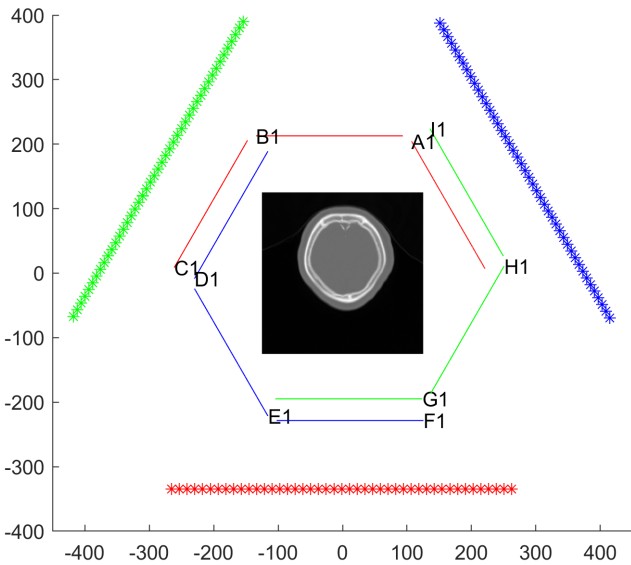

Figure 9: The illustration of s-HCT (Luo et al., 2021) geometry with three linear CNT x-ray source arrays in a triangular shape. The star marks indicate the source. The lines indicate the detectors in three planes that receive the signal from the corresponding source in the same color.

## A.2 QUANTITATIVE RESULTS FOR FWI

Table 4: **Quantitative results for FWI**, compared with InversionNet and Auto-Linear, in terms of MAE, MSE, and SSIM. For each dataset, we use bold to highlight the best results.

| Metrics | Model | FVA | FVB | CVA | CVB | FFA |
|---|---|---|---|---|---|---|
| MAE↓ | HINT (ours) | **0.0037** | **0.0206** | **0.0296** | **0.0894** | **0.0075** |
| | Auto-Linear | 0.0081 | 0.0467 | 0.0738 | 0.1820 | 0.0164 |
| | BigFWI | 0.0055 | 0.0233 | 0.0343 | 0.0933 | 0.0106 |
| MSE↓ | HINT (ours) | **0.0001** | 0.0054 | **0.0059** | **0.0459** | **0.0005** |
| | Auto-Linear | 0.0005 | 0.0151 | 0.0188 | 0.1051 | 0.0026 |
| | BigFWI | **0.0001** | **0.0048** | 0.0064 | 0.0464 | 0.0008 |
| SSIM↑ | HINT (ours) | **0.9965** | **0.9669** | **0.9178** | **0.7963** | **0.9917** |
| | Auto-Linear | 0.9888 | 0.9044 | 0.8057 | 0.6169 | 0.9701 |
| | BigFWI | 0.9943 | 0.9658 | 0.9027 | 0.7808 | 0.9871 |
| Metrics | Model | FFB | CFA | CFB | SA | SB |
| MAE↓ | HINT (ours) | 0.0728 | **0.0143** | 0.1289 | **0.0514** | 0.0558 |
| | Auto-Linear | 0.1208 | 0.0277 | 0.1791 | 0.0719 | 0.0638 |
| | BigFWI | **0.0710** | 0.0167 | **0.1245** | 0.0514 | **0.0553** |
| MSE↓ | HINT (ours) | 0.0186 | **0.0018** | 0.0442 | 0.0081 | 0.0079 |
| | Auto-Linear | 0.0362 | 0.0061 | 0.0697 | 0.0139 | 0.0097 |
| | BigFWI | **0.0175** | 0.0022 | **0.0411** | **0.0075** | **0.0077** |
| SSIM↑ | HINT (ours) | **0.8076** | **0.9784** | **0.6816** | **0.9136** | 0.7532 |
| | Auto-Linear | 0.6868 | 0.9426 | 0.5672 | 0.8423 | 0.7275 |
| | BigFWI | 0.8027 | 0.9712 | 0.6781 | 0.9125 | **0.7567** |

## A.3 VALIDATION ACROSS DIFFERENT RESOLUTIONS

Table 5: Quantitative results with different feature map $z_\psi$ resolutions, in terms of SSIM, evaluated on OpenFWI.

| Metrics | Resolution | FVA | FVB | CVA | CVB | FFA |
|---|---|---|---|---|---|---|
| SSIM↑ | $5 \times 5$ | 0.9957 | **0.9669** | 0.9109 | 0.7925 | 0.9906 |
| | $7 \times 7$ | 0.9958 | 0.9655 | 0.9091 | 0.7882 | 0.9906 |
| | $14 \times 14$ | **0.9965** | **0.9669** | **0.9178** | **0.7963** | **0.9917** |
| | $35 \times 35$ | 0.9948 | 0.9634 | 0.8997 | 0.7733 | 0.9891 |
| Metrics | Resolution | FFB | CFA | CFB | SA | SB |
| SSIM↑ | $5 \times 5$ | 0.7981 | 0.9767 | 0.6727 | 0.9088 | 0.7512 |
| | $7 \times 7$ | 0.7943 | 0.9764 | 0.6702 | 0.9064 | 0.7456 |
| | $14 \times 14$ | **0.8076** | **0.9784** | **0.6816** | **0.9136** | **0.7532** |
| | $35 \times 35$ | 0.7939 | 0.9725 | 0.6777 | 0.9079 | 0.7384 |