# OpenReview forum: "On a Hidden Property in Computational Imaging"
_ICLR.cc/2025/Conference — Submitted to ICLR 2025_

### Official Review · Reviewer_Uz6Q · 2024-10-29

**Soundness:** 3
**Presentation:** 4
**Contribution:** 3
**Rating:** 6
**Confidence:** 3

**Summary:**

This paper present a framework for solving inverse problems like the full waveform inversion. The problem is tackled by projecting the data inside an embedding space which linearize the initial condition. Furthermore the proposed method combined a reconstruction and a inversion target and the two embedding spaces are linked using wave equations. Such result is of large interest for the inverse problem community. The framework shows promising results against to state-of-art methods.

**Strengths:**

The paper presents a very interesting framework for solving a class of inverse problems.

1. **Originality**: while this paper relies on previous recent research, the HINT framework is new. The multi-path FINOLA is clearly a novelty for this problematic. The hidden wave phenomenon is very intriguing and the underling properties seem pretty helpful to inverse.
2. **Quality**: the method is well motivated and the context is clear. The link with previous work is done with the contribution clearly highlighted.
3. **Clarity**: the paper is easy to read and most of the components are described.
4. **Significance**: since inverse problems are an important class of problems in signal, imaging... such work can have a big impact on the community.

**Weaknesses:**

There few weakness is the article, they are minor but they impact my final score.

1. The class of inverse problems that can be considered is unclear. Do we have an idea of which problems involved an hidden wave phenomenon? Even an insight would be welcome.
2. The multi-path FINOLA need a better description. I don't see the "multipath" in the equations.
3. One small experiment to compare the method with classical framework (LASSO with wavelets...) would interesting to have a full idea on the effectiveness of the framework.

**Questions:**

* I have a question on the class of inverse problems, do modalities like MRI or tomography enter in the framework?
* Please clarify the part about the multipath.

---

> ### Author Response · Authors · 2024-11-20
> **Rebuttal by Authors**
>
> **Questions 1: The class of inverse problems that can be considered is unclear. Do we have an idea of which problems involved an hidden wave phenomenon? Even an insight would be welcome. Do modalities like MRI or tomography enter in the framework?**
>
> Thank you for your excellent question. To answer this question definitively, extensive experimentation across all range of computational imaging tasks is required. Works by Chen et al. (2023a, b) demonstrate that the hidden wave phenomenon is held on ImageNet, which encompasses a wide variety of data distributions. This suggests that for single-modality autoregression, the hidden wave property is likely to exist for most image data. However, the critical question is under what circumstances two modalities can share the same wave speeds and exhibit a linear correlation in their initial conditions.
>
> We think that a possible condition for this phenomenon is that the two modalities should represent different views of the same underlying essence, implying a strong correlation between them.
> Additionally, we think the presence of this phenomenon is related to whether the measurement data contains sufficient information about the target property. If the uncertainty in the data is excessively large that the inverse problem is particularly challenging to solve, the hidden wave property is less likely to hold.
>
> In our study, we tested three tasks—FWI, CT, and EM inversion—and observed that the hidden wave phenomenon holds across all of them. These tasks involve vastly different governing equations, which suggests that the hidden wave property has broad applicability across a wide range of computational imaging problems. Since Computed Tomography is already validated within this framework, we believe that MRI, another linear inverse problem, should also fall within this framework. We plan to explore additional scenarios, including MRI and other modalities, in future work to deepen our understanding of the underlying similarities and conditions that give rise to this property.
>
> **Questions 2: The multi-path FINOLA need a better description. I don't see the "multipath" in the equations.**
>
> Apologies for the confusion. The multi-path FINOLA is an implementation method of FINOLA and is not directly tied to the main equations presented in our paper. We will include the following clarification in the text:
>
> "The multi-path FINOLA implementation divides the initial conditions vector into multiple smaller vectors, with each vector processed independently through the FINOLA framework. All paths share the same parameters. The resulting feature maps, each corresponding to a specific solution that satisfies the necessary constraints, are then aggregated to construct the final solution."
>
> **Questions 3: ne small experiment to compare the method with classical framework (LASSO with wavelets...) would interesting to have a full idea on the effectiveness of the framework.**
>
> The classical framework you mentioned, such as LASSO with wavelets, is conceptually similar to our baseline method, SIRT, which is also a widely used classical approach. To provide additional context, we conducted experiments using LASSO with Haar wavelets. The results, shown in the table below, indicate that both SIRT and LASSO with Haar wavelets yield similar performance.
>
> | Model                     | MAE ↓       | MSE ↓       | SSIM ↑       |
> |---------------------------|-------------|-------------|--------------|
> | HINT                      | **31.95**   | 9754.48     | 0.9843       |
> | SIRT                      | 45.67       | 6510.67     | **0.9918**   |
> | LASSO with Haar Wavelet   | 45.94       | **6386.31** | 0.9904       |
>
> ---
> Thank you for dedicating your time and effort to provide feedback on our work.

---

> > ### Comment · Reviewer_Uz6Q · 2024-11-25
> >
> > Thank you for the answer. I have now a better understanding of the idea. I have read the other reviews, I agree that this hidden wave phenomenon should be better addressed. For this reason I will not change my score.

---

> ### Author Response · Authors · 2024-11-23
> **Reminder: Approaching End of Author-Reviewer Discussion Period**
>
> Dear Reviewer Uz6Q,
>
> We hope this message finds you well. We want to extend a friendly reminder that the author-reviewer discussion period is nearing its close. Your input and perspective have been highly valuable throughout this process. If any questions or concerns about our rebuttal arise, we're here to listen and address them in a timely manner. Your insights are crucial in refining our manuscript further.
>
> Understanding the demands of your time, we deeply appreciate your active participation in this rebuttal discussion. Your contributions are invaluable to us, and we genuinely thank you for your dedication.
>
> Should you have any inquiries or thoughts, please don't hesitate to reach out.
>
> Best regards,
>
> -Authors

---

> ### Author Response · Authors · 2024-11-27
> **Reply to comments**
>
> Dear Reviewer Uz6Q,
>
> Thank you for your feedback and for sharing your perspective. We greatly appreciate the time and effort you have devoted to reviewing our manuscript.
>
>
> To further support our validation of the proposed property and deepen our understanding of the latent space structure, we conducted an additional experiment generating a full-resolution feature map (70 $\times$ 70) in the latent space based on our proposed property, with minimal reliance on the decoder. As shown in the following table, this configuration achieves performance consistent with other setups. This result demonstrates that the decoder, with only 0.4M parameters, is dwarfed by the complexity of the linear converter and Finola modules (0.3M + 1.1M parameters). Consequently, the output is primarily governed by the regularizations, which effectively embody the proposed properties.
>
>
> | Metrics      | Resolution    | FVA       | FVB         | CVA       | CVB       | FFA       | FFB       | CFA       | CFB       | SA        | SB        |
> |--------------|---------------|-----------|-------------|-----------|-----------|-----------|-----------|-----------|-----------|-----------|-----------|
> | **SSIM ↑**   | 5 $\times$ 5         | 0.9957    | **0.9669**  | 0.9109    | 0.7925    | 0.9906    | 0.7981    | 0.9767    | 0.6727    | 0.9088    | 0.7512    |
> |              | 7 $\times$ 7         | 0.9958    | 0.9655      | 0.9091    | 0.7882    | 0.9906    | 0.7943    | 0.9764    | 0.6702    | 0.9064    | 0.7456    |
> |              | 14 $\times$ 14       | **0.9965**| **0.9669**  | **0.9178**| **0.7963**| **0.9917**| **0.8076**| **0.9784**| **0.6816**| **0.9136**| **0.7532**|
> |              | 35 $\times$ 35       | 0.9948    | 0.9634      | 0.8997    | 0.7733    | 0.9891    | 0.7939    | 0.9725    | 0.6777    | 0.9079    | 0.7384    |
> |              | 70 $\times$ 70       | 0.9937    | 0.9621      | 0.8984    | 0.7724    | 0.9883    | 0.7927    | 0.9719    | 0.6754    | 0.9063    | 0.7371    |
>
>
>
> While we understand your decision to maintain your score, we would like to kindly ask if you have any comments or suggestions regarding the paper. Your expertise is invaluable to us, and we are committed to improving the quality and clarity of our work. If there are specific aspects of the paper that you believe could be further refined or strengthened, we would be deeply grateful for your insights.
>
> Thank you again for your thoughtful engagement throughout this process. We look forward to hearing any additional thoughts you may have.
>
> Best regards,
>
> The authors

---

### Official Review · Reviewer_bUtp · 2024-11-03

**Soundness:** 2
**Presentation:** 1
**Contribution:** 1
**Rating:** 3
**Confidence:** 2

**Summary:**

The paper considers three inverse problems (namely, FWI, CT reconstruction, and EM inversion). The authors propose a new architecture for the reconstruction operator by exploiting the relationship between the latent representation of the measured data and the parameters to be reconstructed.  In particular, the construction of the reconstruction net exploits the fact that the data and the parameters, in their latent space, are governed by the same wave equation with linearly correlated initial conditions. Numerical experiments are conducted to demonstrate the performance of the new architecture as compared with a couple of baseline methods for the three inverse problems mentioned above.

**Strengths:**

The paper proposes an interesting idea to extend FINOLA (first-order norm+linear autoregressive modeling) to consider both the data space and the parameter space. This can potentially lead to useful architectures for various other inverse problems.

**Weaknesses:**

1. In my opinion, the paper’s main contribution is to propose a new architecture, and not establish any fundamental “hidden property” in computational imaging problems (contrary to what the abstract and the introduction attempt to portray). This also makes the overall presentation somewhat misleading and difficult to follow.

2. The numerical experiments do not provide strong evidence in favor of the proposed method. The baseline methods for comparison (e.g., SIRT and InversionNet for CT) are chosen somewhat arbitrarily. State-of-the-art deep learning methods for CT (such as learned primal-dual by Adler and Oktem) are not used for comparison, making it difficult to judge the empirical superiority of the new architecture.

**Questions:**

The overall exposition is somewhat difficult to follow, primarily because of the lack of clarity about the paper’s main contributions and the usage of non-standard terminologies in comparison with the inverse problems literature. For instance, the word “modality” is used to refer to the parameter and data spaces, which can lead to confusion. Some specific comments are below:

- Abstract: “where both modalities are governed by complex mathematical equations”: This is a rather vague statement to use in the abstract. What specific governing equations are being referred to here? What latent space is talked about here? The actual contributions or the significance of the proposed method do not come out clearly from the abstract.

- Figure 1 caption (and other places in the introduction): The phrase “latent space” is used frequently without much explanation about what exactly it refers to.

- Page 2: “Whether an elegant mathematical relationship exists in the latent space, akin to that
in the original space?”: I don’t think it is a precise, well-formulated research question. The mathematical relationship between the parameter space and the data space is determined by the specific imaging modality, whereas the relationship in the “latent space” is empirically enforced (and no such relationships are shown to exist theoretically).

- Page 2: “...typically with a bottleneck in the network, they lack a deeper understanding of these latent representations.”: I don’t see any such “deeper understanding” (which in itself is somewhat vague and subjective) being uncovered in this paper either.

- The phrase “target property” is used in several places in the paper. Could you please explain what this means?

- Section 2.1: It might be good to make the descriptions of the inverse problems (FWI, CT, and EM inversion) more concise.

- Page 6: “Difference with vanilla FINOLA”: This part needs to be rewritten. Currently, it gives the impression that the proposed method is capable of handling multi-modal data for reconstruction, which it isn’t.

- Page 7: Architecture details: Do you use the same architecture for the reconstruction network for all three inverse problems considered?

- CT experiments: There is no comparison with the unrolling-based techniques (such as learned primal-dual), which are known to yield state-of-the-art reconstruction performance. The choice of the baseline techniques is somewhat arbitrary and not well-motivated.

---

> ### Author Response · Authors · 2024-11-20
> **Rebuttal by Authors**
>
> **Weaknesses: In my opinion, the paper's main contribution is to propose a new architecture, and not establish any fundamental “hidden property” in computational imaging problems. State-of-the-art deep learning methods for CT are not used for comparison, making it difficult to judge the empirical superiority of the new architecture.**
>
> We appreciate your comments and understand the concerns regarding the paper's contributions. We would like to clarify that the primary goal of this work is not to propose a new architecture or achieve state-of-the-art (SOTA) performance. Rather, the focus is to uncover and validate a hidden property in computational imaging problems.
>
> To achieve this goal, we intentionally designed a network with strong spatial regularizations and hard constraints, prioritizing mathematical simplicity and interpretability over raw performance. These strong restrictions enable us to uncover insights and highlight the hidden property we propose. While this framework does not aim for SOTA performance, it serves as a highly regulated framework for revealing the mathematical insights that constitute the core contribution of our work. This focus distinguishes our study from traditional architecture-driven advancements.
>
> As stated in the paper, this work **empirically** demonstrates the hidden property, which is revealed through the strong regularization applied within the network, rather than through theoretical derivation. Analogous to how LASSO regularization reflects the sparsity property of data, we use linear constraints and the FINOLA method to regulate the latent space in alignment with the proposed property. These constraints ensure that the hidden property emerges clearly, facilitating its validation and analysis.
>
> We hope this clarification better aligns our intended contributions with the reviewer's expectations and demonstrates how our work advances the understanding of computational imaging problems by providing novel insights into their underlying structure.
>
> **Questions 1:  The usage of non-standard terminologies in comparison with the inverse problems literature. For instance, the word “modality” is used to refer to the parameter and data spaces, which can lead to confusion.**
>
> Sorry for the confusion. We intended to adopt more machine learning terminologies to serve readers from this venue better. To address this, we will clarify our terminology in the Introduction, page 2, lines 54-56, as follows:
>
> "These methods address inverse problems involving two modalities, referring to the reconstruction of physical properties (parameter space) from measured data (data space), a process governed by linear or nonlinear mathematical equations (Kirsch et al., 2011)."
>
> This revision aims to bridge the terminological gap between the inverse problems literature and the machine learning audience, ensuring clarity while maintaining accessibility for readers from both domains.
>
> **Questions 2:  Figure 1 caption (and other places in the introduction): The phrase “latent space” is used frequently without much explanation about what exactly it refers to.**
>
> The phrase "latent space" is a commonly used concept in machine learning, referring to a lower-dimensional space of latent variables—also known as hidden or unobserved variables. These variables are not directly observable in the physical world but are learned by machine learning algorithms to capture the essential features and underlying structures of the input data.

---

> ### Author Response · Authors · 2024-11-20
> **Rebuttal by Authors**
>
> **Questions 3:  Abstract: “where both modalities are governed by complex mathematical equations”: This is a rather vague statement to use in the abstract. What specific governing equations are being referred to here? What latent space is talked about here? The actual contributions or the significance of the proposed method do not come out clearly from the abstract.**
>
> **Governing Equations:** To clarify, the "complex mathematical equations" referenced in the abstract refer to the governing equations underlying the computational imaging tasks discussed in the paper, such as the acoustic wave equation in FWI, the linear integral equation in CT, and Maxwell's equations in EM inversion. Given the diversity of these tasks, it would be overly redundant to detail each equation in the abstract. Instead, we use general terminology to highlight their commonalities, while providing a detailed review of these equations and tasks in the main text.
>
> **Latent Space:** As explained in Question 2, the latent space represents a lower-dimensional space of latent variables learned by the network to capture the essential features and underlying structures of the input data.
>
> **Contributions:** Furthermore, as stated in our abstract and discussed in Question 1, the primary goal of this work is not to propose a new architecture or achieve SOTA performance. Instead, the focus is on uncovering and validating the mathematical insights of the structure of latent spaces in deep learning models for computational imaging tasks.
>
> **Questions 4:  “Whether an elegant mathematical relationship exists in the latent space, akin to that in the original space?”: I don't think it is a precise, well-formulated research question. The mathematical relationship between the parameter space and the data space is determined by the specific imaging modality, whereas the relationship in the “latent space” is empirically enforced (and no such relationships are shown to exist theoretically).**
>
>
> Recent works (Feng et al., 2022, 2024) have demonstrated that physical properties with strong mathematical relationships (e.g., waveform data and velocity map) exhibit simpler relationships (e.g., linear) when projected into latent space with appropriate encoders. Building on this idea, our work aims to further explore and advance this understanding. Unlike prior studies that relied on complex encoder designs to generate latent spaces, our approach employs autoregression governed by a set of wave equations. This introduces much stronger regularizations on the feature map, reducing the degrees of freedom in both the encoder and decoder and resulting in a more constrained and interpretable latent space structure.
>
> As an experimental verification paper, our focus is on empirical validation rather than theoretical derivation. The validation and ablation experiments confirm that the added regularizations do not degrade performance. Instead, they demonstrate that the latent space, described by these proposed mathematical properties, is naturally contained in the potential optimal solutions of the latent space. This suggests our approach does not force the model into an artificially flawed latent space.
> The key contribution of this work is the discovery and empirical validation of this latent space property. While the findings are empirical, they offer a novel perspective on the mathematical structure of latent representations in computational imaging tasks.
>
>
> [1] Feng, Yinan, et al. "Auto-Linear Phenomenon in Subsurface Imaging." Forty-first International Conference on Machine Learning, 2024.
>
> [2] Feng, Yinan, et al. "An intriguing property of geophysics inversion." Thirty-ninth International Conference on Machine Learning. PMLR, 2022.

---

> ### Author Response · Authors · 2024-11-20
> **Rebuttal by Authors**
>
> **Questions 5: Page 2: “...typically with a bottleneck in the network, they lack a deeper understanding of these latent representations.”: I don't see any such “deeper understanding” (which in itself is somewhat vague and subjective) being uncovered in this paper either.**
>
> In this work, the term "deeper understanding" refers to uncovering and validating the hidden property in the latent space that reflects the relationships of two modalities. While this is not a theoretical derivation, we demonstrate the existence of this property empirically by applying strong spatial regularizations and hard constraints to the network. These constraints simplify the latent space representation, contributing to a novel and deeper understanding of the structure of latent spaces in deep learning models for computational imaging tasks.
>
> **Questions 6: The phrase “target property” is used in several places in the paper. Could you please explain what this means?**
>
>
> The phrase "target property" is a shorthand for "target physical property," referring to the physical properties or parameters that need to be reconstructed or inferred. To address potential confusion, we will clarify this in the Introduction, page 2, line 78, as follows:
>
> "..... we show that the latent spaces of both measurement data and target physical properties (shortened as target properties) are governed ......"
>
> **Questions 7: Section 2.1: It might be good to make the descriptions of the inverse problems (FWI, CT, and EM inversion) more concise.**
>
> Thank you for your suggestion. We will revise Section 2.1 to be more concise as follows:
>
> "
> **Full Waveform Inversion (FWI):**
> FWI is a well-known method to infer subsurface acoustic velocity maps ($c(\mathbf{r})$) from seismic waveform data ($p(\mathbf{r},t)$), collected via sensors during seismic surveys. $\mathbf{r}=(x,y)$ represents spatial location coordinates, and $t$ denotes time. Each receiver logs a 1D time series signal, and the collective signals from all receivers form the waveform data. Given the source term $s(\mathbf{r}, t)$ and the Laplacian operator $\nabla^2$, the process is governed by the acoustic wave equation:
>
> $$
> \nabla^2p(\mathbf{r},t) -\frac{1}{c^2(\mathbf{r})} \frac{\partial^2}{\partial t^2}p(\mathbf{r},t) = s(\mathbf{r},t).
> $$
>
> In this problem, the target property $\mathbf{\psi}$ is the velocity map $c(\mathbf{r})$, and the measurement data $\mathbf{P}$ is the waveform data collected by surface sensors (i.e., $z=0$), abbreviated as $p(x, t)$.
>
> ---
>
> **Computed Tomography (CT):**
> CT is a vital imaging technique to reconstruct an object's internal structure ($f(x, y)$) from projection data ($p(\mathbf{d}, \mathbf{s})$) of X-rays passing through the object, where $(x, y)$ are the spatial coordinates, $\mathbf{s}=(x_s, y_s)$ is the X-ray source position, and $\mathbf{d} = (x_d, y_d)$ is the detector position. Mathematically, the projection data is the integral of the attenuation coefficient $f(x, y)$ along the line segment $L(\mathbf{s}, \mathbf{d})$ connecting the source $\mathbf{s}$ and the detector $\mathbf{d}$:
>
> $$
> p(\mathbf{d}, \mathbf{s}) = \int_{L(\mathbf{s}, \mathbf{d})} f(x, y) \, ds.
> $$
>
> In this problem, the target property $\mathbf{\psi}$ is the attenuation image $f(x, y)$, and the measurement data $\mathbf{P}$ is the projection data $p(\mathbf{d}, \mathbf{s})$.
>
> ---
>
> **Electromagnetic (EM) Inversion:**
> EM inversion focuses on recovering subsurface conductivity ($\sigma$) from electromagnetic field measurements ($\mathbf{E}$ and $\mathbf{H}$). Let $\mathbf{J}$ and $\mathbf{P}$ be the electric and magnetic sources, and $\mu_{0}$ be the magnetic permeability of free space. The governing equations are the time-harmonic Maxwell's Equations:
>
> $$
> \sigma\mathbf{E} - \nabla\times\mathbf{H} = -\mathbf{J},
> $$
>
> $$
> \nabla\times\mathbf{E} + i\omega\mu_{0}\mathbf{H} = -\mathbf{M}.
> $$
>
> In this problem, the target property $\mathbf{\psi}$ is the electrical conductivity $\sigma$, and the measurement data $\mathbf{P}$ is the electric and magnetic fields $\mathbf{E}$ and $\mathbf{H}$.
>
> "

---

> ### Author Response · Authors · 2024-11-20
> **Rebuttal by Authors**
>
> **Questions 8: Page 6: “Difference with vanilla FINOLA”: This part needs to be rewritten. Currently, it gives the impression that the proposed method is capable of handling multi-modal data for reconstruction, which it isn't.**
>
> We extend FINOLA to handle two modalities in computational imaging: measurement data reconstruction and target physical property inference simultaneously. We have revised the text to clarify that the concept of "two modalities" used here differs from the conventional meaning of multi-modal data in computer vision.
>
> "
> **Difference with vanilla FINOLA:**
> Unlike vanilla FINOLA, which is designed for single-modality autoregression, our method extends to two modalities by sharing parameters across both domains. While vanilla FINOLA focuses on modeling single-domain data invariants, we adapt it for computational imaging tasks by enabling the joint representation of measurement data and target physical properties. This joint representation captures the relationship between the two domains, governed by wave equations, offering a unified framework for their relationship. It is important to note that this does not involve handling multi-modal data in the conventional sense but specifically to handle two modalities in computational imaging: measurement data reconstruction and target physical property inference.
> "
>
> **Questions 9: Page 7: Architecture details: Do you use the same architecture for the reconstruction network for all three inverse problems considered?**
>
> Yes, the same architecture is used for the reconstruction network across all three inverse problems. However, due to differences in the data sizes for each problem, the network size and hyperparameters are adjusted accordingly.
>
> ---
>
> Thank you for dedicating your time and effort to provide feedback on our work.

---

> ### Author Response · Authors · 2024-11-23
> **Reminder: Approaching End of Author-Reviewer Discussion Period**
>
> Dear Reviewer bUtp,
>
> We hope this message finds you well. We want to extend a friendly reminder that the author-reviewer discussion period is nearing its close. Your input and perspective have been highly valuable throughout this process. If any questions or concerns about our rebuttal arise, we're here to listen and address them in a timely manner. Your insights are crucial in refining our manuscript further.
>
> Understanding the demands of your time, we deeply appreciate your active participation in this rebuttal discussion. Your contributions are invaluable to us, and we genuinely thank you for your dedication.
>
> Should you have any inquiries or thoughts, please don't hesitate to reach out.
>
> Best regards,
>
> -Authors

---

> ### Comment · Reviewer_bUtp · 2024-11-24
>
> The response appears self-contradictory to me. While the authors claim that "the primary goal of this work is not to propose a new architecture or achieve state-of-the-art (SOTA) performance", they mention in the very next paragraph that they "intentionally designed a network with strong spatial regularizations and hard constraints". The latter is what I understood from the paper, and that is essentially equivalent to proposing a new model for the reconstruction network (i.e., an architecture that incorporates these constraints). I am still of the opinion that the claims of the paper in the abstract (and the introduction) are stronger than the actual contributions (and mildly misleading). The claims of superior interpretability and "deep understanding" of the latent representations are also not substantiated enough.
>
> Overall, I am of the opinion that the theoretical contributions of the paper are limited to none, while the empirical performance is also underwhelming. I would, therefore, like to keep my original evaluation score as is.

---

> ### Author Response · Authors · 2024-11-27
> **Reply to comments**
>
> ## Reply to comments
>
> **On the Network Design and Its Purpose.**
> While it is true that we intentionally designed a network with strong spatial regularizations and hard constraints, this design does not contradict our claim that the primary goal of this work is not to propose a new architecture or achieve state-of-the-art (SOTA) performance. *The architecture itself is not the focus; rather, it is a means to validate the proposed property effectively.*
>
> The purpose of the architecture is to translate the proposed property from mathematical equations into practical implementation, thereby empirically validating the existence of the hidden property we proposed. The network acts as a tool to impose the constraints necessary for revealing this property and is not intended as a general-purpose reconstruction model for a state-of-the-art solution.
>
> To further support our claim that the architecture is designed to validate the proposed property and deepen our understanding of the latent space structure, rather than to achieve SOTA, we conducted an additional experiment generating a full-resolution feature map (70 $\times$ 70) in the latent space based on our proposed property, with minimal reliance on the decoder. As shown in the following table, this configuration achieves performance consistent with other setups. This result demonstrates that the decoder, with only 0.4M parameters, is dwarfed by the complexity of the linear converter and Finola modules (0.3M + 1.1M parameters). Consequently, the output is primarily governed by the regularizations, which effectively embody the proposed properties.
>
>
> | Metrics      | Resolution    | FVA       | FVB         | CVA       | CVB       | FFA       | FFB       | CFA       | CFB       | SA        | SB        |
> |--------------|---------------|-----------|-------------|-----------|-----------|-----------|-----------|-----------|-----------|-----------|-----------|
> | **SSIM ↑**   | 5 $\times$ 5         | 0.9957    | **0.9669**  | 0.9109    | 0.7925    | 0.9906    | 0.7981    | 0.9767    | 0.6727    | 0.9088    | 0.7512    |
> |              | 7 $\times$ 7         | 0.9958    | 0.9655      | 0.9091    | 0.7882    | 0.9906    | 0.7943    | 0.9764    | 0.6702    | 0.9064    | 0.7456    |
> |              | 14 $\times$ 14       | **0.9965**| **0.9669**  | **0.9178**| **0.7963**| **0.9917**| **0.8076**| **0.9784**| **0.6816**| **0.9136**| **0.7532**|
> |              | 35 $\times$ 35       | 0.9948    | 0.9634      | 0.8997    | 0.7733    | 0.9891    | 0.7939    | 0.9725    | 0.6777    | 0.9079    | 0.7384    |
> |              | 70 $\times$ 70       | 0.9937    | 0.9621      | 0.8984    | 0.7724    | 0.9883    | 0.7927    | 0.9719    | 0.6754    | 0.9063    | 0.7371    |
>
>
>
> **Theoretical and Experimental Contributions.**
> We acknowledge that this work does not make theoretical contributions in the traditional sense. However, the focus of this paper is on experimental verification, which we believe is equally important for advancing understanding in this domain. The strong regularizations and constraints in the network are a specific embodiment and implementation of the proposed property, allowing us to empirically demonstrate its existence. This experimental approach complements theoretical work and provides a foundation for future investigations.

---

> > ### Comment · Reviewer_bUtp · 2024-11-27
> >
> > Thanks for the clarification. I respectfully disagree with the authors regarding the quantum of novelty in the paper.

---

> ### Author Response · Authors · 2024-11-27
> **Reply to comments**
>
> Dear Reviewer bUtp,
>
> Thank you for your feedback; we genuinely appreciate your candid perspective. We understand that you disagree with our assessment of the novelty in the paper. Could you kindly elaborate on the specific aspects you disagree with? Or do you disagree with every aspect? Your clarification would be invaluable in helping us address your concerns and improve our work.
>
> Best regards,
>
> The authors

---

> > ### Comment · Reviewer_bUtp · 2024-11-28
> >
> > I understand and appreciate the authors' argument about the empirical contributions of the paper (and their practical relevance for some inverse problems). Nonetheless, the lack of theoretical contributions makes the paper less appealing in my view. I have kept my confidence score low so that the other reviewers and the AC (who might be in a better position to assess the paper's empirical contribution) can judge whether it merits publication in ICLR.

---

### Official Review · Reviewer_pMMZ · 2024-11-04

**Soundness:** 2
**Presentation:** 2
**Contribution:** 2
**Rating:** 5
**Confidence:** 2

**Summary:**

This paper considers a hidden property in computational imaging, demonstrated across Full Waveform Inversion (FWI), Computed Tomography (CT), and Electromagnetic (EM) inversion tasks, which reveals that they share a common set of one-way wave equations in the latent space. The authors leverage understanding of this shared latent representation to achieve accurate reconstructions and predictions across imaging tasks, achieving similar or better performance than existing methods but with fewer parameters.

**Strengths:**

- Results of experiments on computational imaging tasks show simliar or improved performance with fewer model parameters.

**Weaknesses:**

- The work draws very heavily on two prior works by Chen et al. 2023 (a,b).  As far as I can tell neither of these works have been accepted by peer-review venues.
- There is no theoretical motivation for the hidden wave equations, as far as I can tell, although I did not review the cited papers.

**Questions:**

- Can the authors speculate about similarity between the computational imaging tasks considered that might give rise to the observed phenomenon?  Or do they believe this phenomenon should existing for all computational imaging tasks?

---

> ### Author Response · Authors · 2024-11-20
> **Rebuttal by Authors**
>
> **Weaknesses 1: The work draws very heavily on two prior works by Chen et al. 2023 (a,b). As far as I can tell neither of these works have been accepted by peer-review venues.**
>
> Although the works by Chen et al. (2023 a, b) have not yet been accepted by peer-reviewed venues, we find their ideas to be both innovative and interesting. To ensure the reliability and validity of their methods, we reproduced their results on ImageNet before apply them. Moreover, this framework performs well in our specific problem domain. We believe that interesting and reproducible research, regardless of its publication status, can provide valuable insights and serve as a solid foundation for further exploration.
>
> **Weaknesses 2: There is no theoretical motivation for the hidden wave equations, as far as I can tell, although I did not review the cited papers.**
>
> The hidden wave equations provide FINOLA, an autoregressive method, with a better mathematical description without affecting its actual operation. In essence, FINOLA is equivalent to the formulation of the hidden wave equations. It is important to note that empirical results drove the development process—FINOLA was initially proposed and demonstrated to work effectively, which then motivated the authors to formalize its underlying principles mathematically. Thus, the theoretical description arose as a retrospective effort to better articulate the method's foundations, rather than serving as its original motivation.
>
> **Questions: Can the authors speculate about similarity between the computational imaging tasks considered that might give rise to the observed phenomenon? Or do they believe this phenomenon should existing for all computational imaging tasks?**
>
> Thank you for your excellent question. To answer this question definitively, extensive experimentation across all range of computational imaging tasks is required. Works by Chen et al. (2023a, b) demonstrate that the hidden wave phenomenon is held on ImageNet, which encompasses a wide variety of data distributions. This suggests that for single-modality autoregression, the hidden wave property is likely to exist for most image data. However, the critical question is under what circumstances two modalities can share the same wave speeds and exhibit a linear correlation in their initial conditions.
>
> We think that a possible condition for this phenomenon is that the two modalities should represent different views of the same underlying essence, implying a strong correlation between them.
> Additionally, we think the presence of this phenomenon is related to whether the measurement data contains sufficient information about the target property. If the uncertainty in the data is excessively large that the inverse problem is particularly challenging to solve, the hidden wave property is less likely to hold.
>
> In our study, we tested three tasks—FWI, CT, and EM inversion—and observed that the phenomenon holds across all of them. These tasks involve vastly different governing equations, which suggests that the hidden wave property should be applicable across a wide range of computational imaging tasks. However, whether it generalizes to all such tasks remains an open question that requires further investigation. We plan to explore more scenarios in future work to deepen our understanding of the underlying similarities and conditions that give rise to this property.
>
> ---
> Thank you for dedicating your time and effort to provide feedback on our work.

---

> ### Author Response · Authors · 2024-11-23
> **Reminder: Approaching End of Author-Reviewer Discussion Period**
>
> Dear Reviewer pMMZ,
>
> We hope this message finds you well. We want to extend a friendly reminder that the author-reviewer discussion period is nearing its close. Your input and perspective have been highly valuable throughout this process. If any questions or concerns about our rebuttal arise, we're here to listen and address them in a timely manner. Your insights are crucial in refining our manuscript further.
>
> Understanding the demands of your time, we deeply appreciate your active participation in this rebuttal discussion. Your contributions are invaluable to us, and we genuinely thank you for your dedication.
>
> Should you have any inquiries or thoughts, please don't hesitate to reach out.
>
> Best regards,
>
> -Authors

---

> > ### Comment · Reviewer_pMMZ · 2024-11-25
> >
> > Thank you to the authors for responding to my queries.  I have reviewed their response and also the comments by the other reviewers and the authors' corresponding responses.  My original assessment and rating for the paper stands.

---

> ### Author Response · Authors · 2024-11-27
> **Reply to comments**
>
> Dear Reviewer pMMZ,
>
> Thank you for your feedback and for sharing your perspective. We greatly appreciate the time and effort you have devoted to reviewing our manuscript.
>
>
> To further support our validation of the proposed property and deepen our understanding of the latent space structure, we conducted an additional experiment generating a full-resolution feature map (70 $\times$ 70) in the latent space based on our proposed property, with minimal reliance on the decoder. As shown in the following table, this configuration achieves performance consistent with other setups. This result demonstrates that the decoder, with only 0.4M parameters, is dwarfed by the complexity of the linear converter and Finola modules (0.3M + 1.1M parameters). Consequently, the output is primarily governed by the regularizations, which effectively embody the proposed properties.
>
>
> | Metrics      | Resolution    | FVA       | FVB         | CVA       | CVB       | FFA       | FFB       | CFA       | CFB       | SA        | SB        |
> |--------------|---------------|-----------|-------------|-----------|-----------|-----------|-----------|-----------|-----------|-----------|-----------|
> | **SSIM ↑**   | 5 $\times$ 5         | 0.9957    | **0.9669**  | 0.9109    | 0.7925    | 0.9906    | 0.7981    | 0.9767    | 0.6727    | 0.9088    | 0.7512    |
> |              | 7 $\times$ 7         | 0.9958    | 0.9655      | 0.9091    | 0.7882    | 0.9906    | 0.7943    | 0.9764    | 0.6702    | 0.9064    | 0.7456    |
> |              | 14 $\times$ 14       | **0.9965**| **0.9669**  | **0.9178**| **0.7963**| **0.9917**| **0.8076**| **0.9784**| **0.6816**| **0.9136**| **0.7532**|
> |              | 35 $\times$ 35       | 0.9948    | 0.9634      | 0.8997    | 0.7733    | 0.9891    | 0.7939    | 0.9725    | 0.6777    | 0.9079    | 0.7384    |
> |              | 70 $\times$ 70       | 0.9937    | 0.9621      | 0.8984    | 0.7724    | 0.9883    | 0.7927    | 0.9719    | 0.6754    | 0.9063    | 0.7371    |
>
>
>
> While we understand your decision to maintain your score, we would like to kindly ask if you have any comments or suggestions regarding the paper. Your expertise is invaluable to us, and we are committed to improving the quality and clarity of our work. If there are specific aspects of the paper that you believe could be further refined or strengthened, we would be deeply grateful for your insights.
>
> Thank you again for your thoughtful engagement throughout this process. We look forward to hearing any additional thoughts you may have.
>
> Best regards,
>
> The authors

---

### Meta-Review · Area_Chair_92y5 · 2024-12-20

**Metareview:**

This paper explores a "hidden property" in the latent spaces of encoder-decoder architectures for computational imaging, where both unknown parameters and observed data satisfy multidimensional one-way wave equations with correlated initial conditions.

While the idea is intriguing, reviewers found the claims in the abstract and introduction overstated and unsubstantiated, with a lack of clarity on whether the focus is on a hidden property or a new architecture/loss (pMMZ). The paper often reads as a series of vague and arbitrary statements. Additionally, bUtp highlighted insufficient novelty and the absence of a general description of the problem class where this property appears.

The claims of enhanced interpretability are overly strong, as there is no theoretical analysis or broad understanding offered, and the results are tied exclusively to the FINOLA framework, which itself has faced criticism and remains unpublished.

During discussions, the authors stated that the work is about experimentally supporting a "hidden property" rather than trying to establish a causal, mechanistic theory. But then it remains unclear how it contributes to interpretability. Such claims in critical areas like interpretability and trustworthiness require careful justification.

In conclusion, limitations to the FINOLA framework, weak theoretical motivation (92y5), discrepancies between claims and results, issues with novelty and experiments (bUtp), and the absence of a clear problem class or mechanism (92y5, Uz6Q) compel me to recommend rejection.

**Additional Comments On Reviewer Discussion:**

The reviewers asked concrete questions about the scope of the method and criticized clarity, vagueness, reliance on FINOLA, ... . The authors' response that the difference from FINOLA is that they extend it to two modalities (by which they mean objects and measurements, sometimes referred to as "parameters and data" in "real" inverse problems literature; I agree with the reviewers that calling these modalities is strange). This is unfortunately not sufficient for acceptance to ICLR, even with everything else in order. In the end, the reviewers did not change their critical opinion. My recommendation is based on a careful reading of the discussions, the paper, and the context of prior art. There is not enough novelty nor clarity for acceptance.

---

### Decision · Program_Chairs · 2025-01-22

Reject